# Change of surfactant protein D and A after renal ischemia reperfusion injury

**Islam Md Imtiazul[1], Redwan Asma[1], Ji-Hye Lee[2], Nam-Jun Cho[3], Samel Park[3], Ho-Yeon Song[1], Hyo-Wook Gil[3]***

**1** Department of Microbiology, College of Medicine, Soonchunhyang University, Cheonan, Republic of Korea,
**2** Department of Pathology, Soonchunhyang University Cheonan Hospital, Cheonan, Republic of Korea,
**3** Department of Internal Medicine, Soonchunhyang University Cheonan Hospital, Cheonan, Republic of Korea

* hwgil@schmc.ac.kr

**Data Availability Statement:** All relevant data are within the manuscript and its Supporting Information files.

**Funding:** This research was supported by a grant of the Korea Health Technology R&D Project

## Abstract

Acute kidney injury (AKI) is associated with widespread effects on distant organs, including the lungs. Surfactant protein (SP)-A and SP-D are members of the C-type lectin family, which plays a critical role in host defense and regulation of inflammation in a variety of infections. Serum levels of SP-A and SP-D are markers to reflect lung injury in acute respiratory distress syndrome, idiopathic pulmonary fibrosis, and sarcoidosis. We investigated the change of lung-specific markers, including SP-A and SP-D in an AKI mice model. We studied C57BL/6J mice 4 and 24 hours after an episode of ischemic AKI (23 min of renal pedicle clamping and then reperfusion); numerous derangements were present, including SP-A, SP-D, and lung tight-junction protein. Neutrophil infiltration and apoptosis in the lungs increased in ischemic AKI. Receptor for advanced glycation end products (RAGE) in the lungs, a marker of pneumocyte I, was not changed. Lung tight-junction proteins, particularly claudin-4, claudin-18, and anti-junctional adhesion molecule 1 (JAMA-1), were reduced in 24 hours after AKI. Serum SP-A and SP-D significantly increased in ischemic AKI. SP-A and SP-D in the lungs did not increase in ischemic AKI. The immunohistochemistry showed that the expression of SP-A and SP-D was intact in ischemic AKI. SP-A and SP-D in the kidneys were significantly higher in AKI than in the sham. These patterns of SP-A and SP-D in the kidneys were similar to those of serum. AKI induces apoptosis and inflammation in the lungs. Serum SP-A and SP-D increased in ischemic AKI, but these could have originated from the kidneys. So serum SP-A and SP-D could not reflect lung injury in AKI. Further study is needed to reveal how a change in lung tight-junction protein could influence the prognosis in patients with AKI.

## Introduction

Acute kidney injury (AKI) is associated with high mortality and morbidity [1–3]. Some reports suggest that AKI is a systemic disease that adversely affects other organs including bone, gut, brain, heart, and lungs [4–7]. The crosstalk between the kidneys and other organs could

through the Korea Health Industry Development Institute (KHIDI), funded by the Ministry of Health & Welfare, Republic of Korea (HI17C-2059-010017) and Soonchunhyang University Research Fund. The funders had no role in study design, data collection and analysis, decision to publish, or preparation of the manuscript.

**Competing interests:** The authors have declared that no competing interests exist.

contribute to the high mortality [8]. Human and experimental animal data support that AKI adversely affects the lungs [9]. The respiratory complication in patients with AKI could develop grave outcomes. Experimental animal study could reveal that AKI itself affects endothelial cell injury because of inflammation and apoptosis in the lungs [9]. Cytokines could be an important mediator to connect the kidneys to the lungs [10, 11]. Recently, it has been reported that early peritoneal dialysis reduces lung inflammation in mice with ischemic acute kidney injury [12]. Revealing and understanding the mechanism of AKI-induced lung injury could improve the outcome in patients with AKI.

Surfactant proteins are mainly produced in type 2 pneumocytes. Surfactant contains four associated proteins, surfactant protein (SP)-A, SP-B, SP-C and SP-D. SP-A and SP-D are members of the C-type lectin family, which plays a critical role in host defense and regulation of inflammation in a variety of infections [13]. Among lung-specific markers, serum levels of SP-A, SP-B, and SP-D have been reported to be increased in acute respiratory distress syndrome [14, 15]. In patients with idiopathic pulmonary fibrosis and sarcoidosis, serum SP-D could predict the extent of parenchymal disease and their survival possibilities [16].

Our hypothesis is that AKI itself could affect the alveolar epithelial cells, causing damage to pneumocytes I and II, which could change the barrier with the change of tight function protein. Therefore, we investigated the change of lung-specific markers including SP-A and SP-D in an AKI mice model.

## Materials and methods

### Animals

We used 8- to 10-week-old male C57BL/6J mice (all from Korea), weighing 20 to 25 g. All experiments were conducted with adherence to the National Institutes of Health Guide for the Care and Use of Laboratory Animals. The animal protocol was approved by the Animal Care and Use Committee of Soonchunhyang University.

### Surgical protocol

Ischemic AKI and a sham operation were performed as previously described [1]. Also, blood was collected and processed as previously described [17], as detailed in S1 Appendix.

Blood was collected by cardiac puncture 4 and 24 hours after the procedure during the sacrificing of the mice by spine dislocation, as detailed in S1 Appendix.

### Collection and preparation of BAL samples

After the collection of blood, the trachea of the mice was dissected and cannulated with a 20-G catheter. Lungs were lavaged with 1 ml of phospahte buffer saline (Gibco, USA, pH 7.4) five times. Samples with <80% return were discarded as the BAL samples that are obtained from <80% return of lavage cannot exactly represent the whole lung status. BAL fluid was centrifuged at 1400×g and 4°C for 5 min, and collected the supernatant and finally stored at -70°C for further use.

### Blood urea nitrogen and serum creatinine measurement

Blood urea nitrogen (BUN) was measured by a specific quantitative colorimetric assay (Quantichrome Urea Assay Kit) from the BioAssay Systems according to the manufacturer's protocol [18]. Serum creatinine was measured by another quantitative colorimetric assay (Creatinine Reagent Set) from POINTE SCIENTIFIC following the manufacturer's instructions.

## Cytokine measurement and Surfactant protein measurements

Preparation of lung lysate sample for ELIA and immunoblot were described in S1 Appendix. Interleukin (IL)-6, tumor necrosis factor alpha (TNF-α), monocyte chemoattractant protein-1 (MCP-1), and receptor for advanced glycation end products (RAGE) level were measured in serum by ELISA using the Mouse IL-6 Quantikine ELISA kit, Mouse TNF-α Quantikine ELISA kit, Mouse/Rat CCL2/JE/MCP-1 Quantikine ELISA Kit, and Mouse RAGE Quantikine ELISA Kit (all from R&D Systems, Inc., Minneapolis, MN, USA) following the manufacturer's instructions.

SP-A and SP-D were determined from BAL, lung lysate, the kidneys and serum by ELISA. Mouse Sftpa1 ELISA Kit (Aviva Systems Biology) and Mouse SP-D Quantikine ELISA Kit (R&D systems, Minneapolis, MN) were used following the manufacturer's instructions to measure SP-A and SP-D respectively.

## Immunoblotting analyses of lung tissue

We did western blot analysis as previously described [16], as detailed in S1 Appendix.

The primary antibodies used in our study included: BCL-2-associated X protein (Bax; Cell Signaling Technology, Danvers, MA) and B cell leukemia/lymphoma 2 (Bcl-2; Cell Signaling Technology), Anti-Claudin 3, Anti-Claudin 4, Anti-Claudin 18, Anti-Junctional Adhesion Molecule 1 (JAMA-1) and Anti-beta Actin all from Abcam.

## Histological detection of kidney tubular injury and lung injury

We did histopathological assays and immunohistochemical assays following the previously described techniques [19]. In brief, the kidney and lung tissues were embedded in paraffin, sliced into 5-μm thick sections, and stained with routine hematoxylin-eosin and Periodic acid–Schiff (PAS) stain. We scored the tubular damage markers by calculating the percentage of tubules that displayed dilatation, desquamation, vacuolization, necrosis, atrophy, casts, interstitial inflammatory cell infiltration, or edema, as follows: 0, none; 1, $\leq$ 10%; 2, 11–25%; 3, 26–50%; 4, 51–75%; and 5, $\geq$ 76%. A semiquantitative assessment of lung infiltration score was carried out according to the following criteria: 0 (rare neutrophils in capillary lumen), 1 (frequent neutrophils in capillary lumen), 2 (extravasation of neutrophils), 3 (aggregation of neutrophils in alveolar wall).

For immunohistochemical assays, SP-A and SP-D (1:1000; Cell Signaling Technology), primary antibodies and anti-rabbit IgG secondary antibody conjugated with biotin (1:1000; LSAB2 kit, Dako-Agilent Technologies, Germany) were used and performed following the protocol provided by Cell Signaling Technology and finally, representing SP-A and SP-D were evaluated using an optical microscope (400×; Carl Zeiss, model Scope.A1).

## Statistical analysis

All experiments were done in triplicate. We did statistical analyses using GraphPad Prism 7 software (GraphPad Software Inc., La Jolla, CA). The differences between AKI and sham were compared using an unpaired Student's *t*-test.

# Results

## Reduced renal functional and histological damages after ischemic AKI

Serum creatinine levels and urea nitrogen were increased after ischemic AKI compared to the sham. The increase showed a time-dependent trend, since the serum creatinine levels and urea nitrogen at 24 hours (AKI model) were higher than were those at 4 hours (AKI model) (Fig

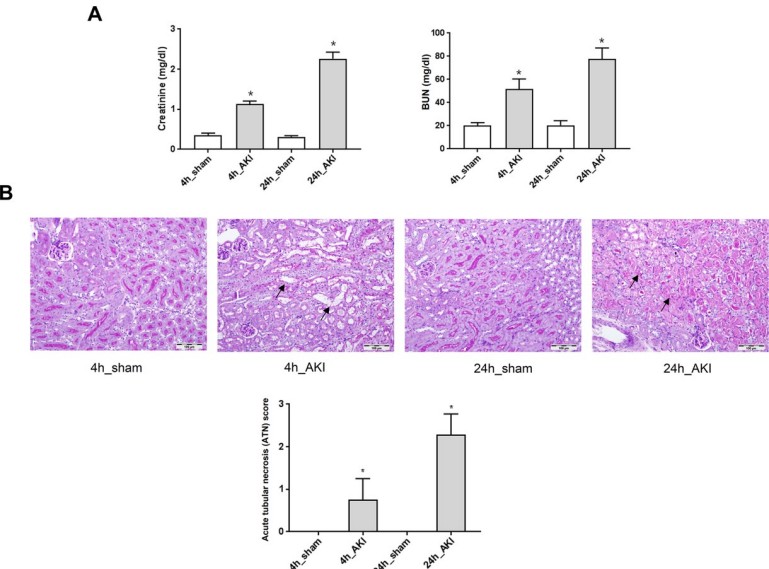

**Fig 1. Change of kidneys in ischemic AKI.** (A) Renal function in C57BL/6 mice renal ischemia reperfusion creatinine and blood urea nitrogen (BUN) evaluated. (B) Kidney sections were subjected to PAS-staining and histological changes were scored. ATN injury scores for PAS-stained kidney sections showed increased tubular injury in ischemic AKI. $^*p < 0.05$ at each time point compared to control.

1A). In addition, the histological examination of the kidney sections indicated greater tubular damage in the kidneys of AKI mice than in those of the sham. The quantitative analysis of tubular injury also showed a higher acute tubular necrosis (ATN) scores for AKI than for the sham. On the other hand, the 24-hour AKI model had a significantly higher rate of kidney damage than did the 4-hour AKI model (Fig 1B).

## Increase of serum cytokines after ischemic AKI

Changes of three different cytokines—IL-6, TNF-α, and MCP-1—were measured in the serum of both the AKI and the sham models after renal ischemia reperfusion. The results showed that the amount of all three cytokines (IL-6, TNF-α, and MCP-1) extremely increased in the serum of the AKI model, whereas the amount was significantly low in the sham (Fig 2).

## Apoptosis and degradation of lung after ischemic AKI

The lung histology showed a significant increase of neutrophil infiltration after ischemic AKI (Fig 3A). AKI-induced apoptosis was found in the lungs of AKI mice. BAX and BCL-2 immunoblot results for the lungs showed that the Bax/Bcl-2 ratio was significantly higher in AKI

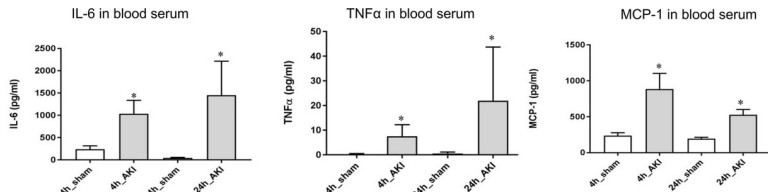

**Fig 2. Changes of serum cytokine in ischemic AKI.** Inflammatory cytokines, interleukin (IL)-6, tumor necrosis factor alpha (TNF-α) by protein, and monocyte chemoattractant protein-1 (MCP-1) were measured. $^*p < 0.05$ at each time point compared to control.

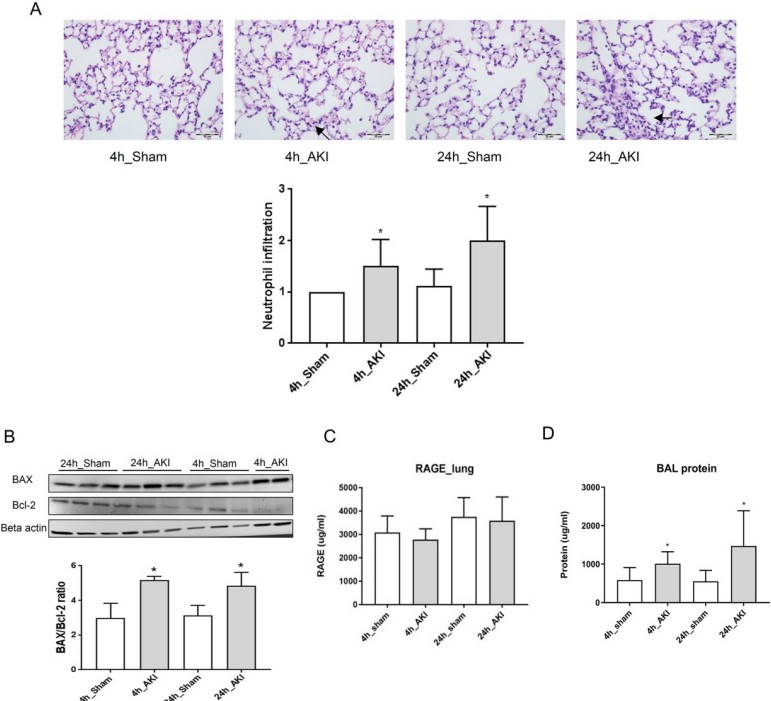

**Fig 3. Change of the lungs in ischemic AKI.** (A) Neutrophil infiltration in the lungs was evaluated in ischemic AKI (S1 Fig). **(B)** Immunoblot of BCL-2-associated X protein (Bax), and B cell leukemia/lymphoma 2 (Bcl-2) were measured. Bax/Bcl-2 ratio in the lungs was identified in ischemic AKI. (C) Receptor for advanced glycation end products (RAGE), marker to reflect pneumocyte I injury in ischemic AKI. (D) Total protein changes in BAL in ischemic AKI. $^*p < 0.05$ at each time point compared to control.

than in the sham (Fig 3B). RAGE in the lungs, a marker of pneumocytes, was not changed after ischemia reperfusion (Fig 3C). However total protein in BAL increased in the AKI model compared with the sham.

According to the immunoblot results, the expression of different tight-junction proteins, particularly claudin-4, claudin-18, and JAMA-1, was reduced in the 24-hour AKI model compared to the sham, but the expression of claudin-3 was similar in both models (Fig 4).

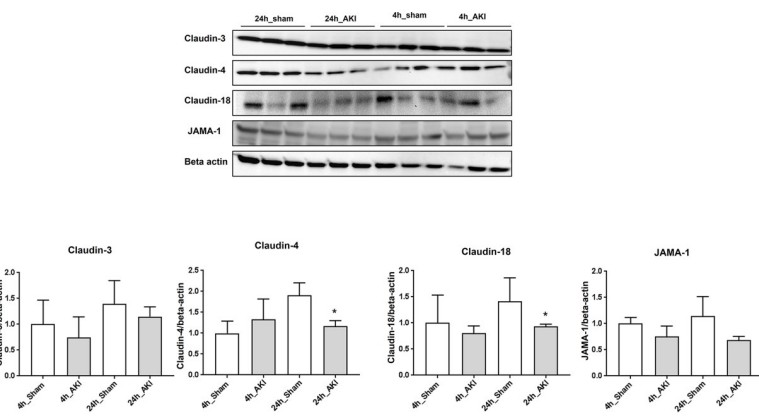

**Fig 4. Change of lung tight-junction protein in ischemic AKI.** Alveolar permeability barrier tight-junction proteins were measured by immunoblot. Claudin-4, claudin-18, and JAMA-1. $^*p < 0.05$ at each time point compared to control.

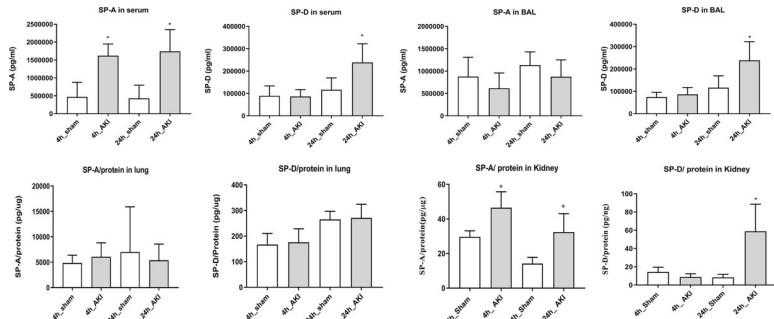

**Fig 5. Change of SP-A and SP-D in ischemic AKI.** SP-A and SP-D in serum changed significantly in ischemic AKI. SP-A and SP-D in the lungs showed no changes in ischemic AKI. SP-A and SP-D in the kidneys were significantly different in AKI than in the sham. *$p < 0.05$ at each time points compared to control.

## Different types of change of surfactant protein D and A after ischemic AKI

Serum level of SP-A significantly increased in 4- and 24-hour AKI compared to each sham. The serum level of SP-D significantly increased in the 24- hour AKI. The BAL level of SP-D was significantly higher in the 24-hour AKI, but SP-A in BAL showed no significant changes. The SP-A and SP-D in the lungs showed no significant differences between AKI and sham (Fig 5). The immunohistochemistry showed the presence of SP-A and SP-D in the lung tissue of AKI and sham, and these protein expressions were intact in the 4- and 24-hour AKI (Fig 6). These finding suggest that pneumocyte II is intact. SP-A and SP-D in the kidneys were significantly higher in AKI than in the sham. These patterns of SP-A and SP-D in the kidneys were similar to those of serum.

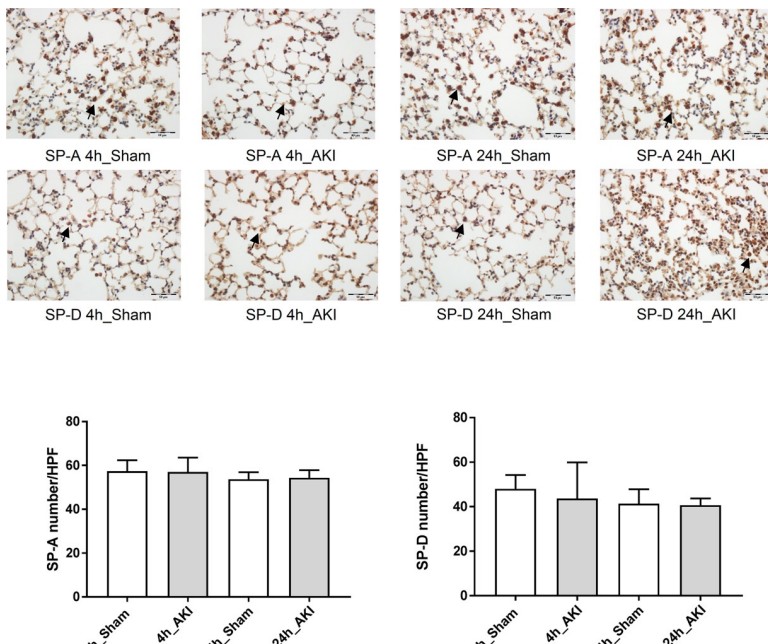

**Fig 6. Presence of SP-A and SP-D in the lung of sham and AKI.** Immunohistochemical localization of SP-A and SP-D in experimental animal lung. The immunohistochemistry showed the expression of SP-A and SP-D in ischemic AKI.

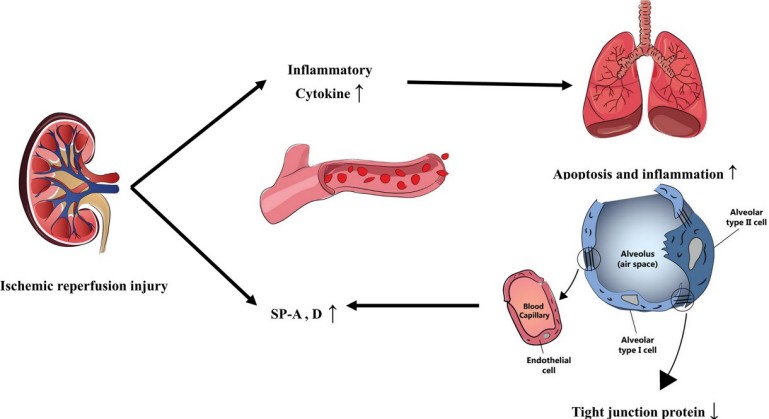

**Fig 7. Schematic of AKI and lung crosstalk.**

## Discussion

The mortality of AKI-associated acute lung injury (ALI) is still high, but the main mechanism is not clear yet [2, 3]. ALI is essentially a noncardiogenic pulmonary edema that occurs in the context of increased alveolar fluid secondary to an increase in lung endothelial or epithelial permeability and/or a decrease in the efficiency of clearance of interstitial fluid [1, 4]. Clinically, the mechanism of lung injury after acute renal injury is considered to be important for improving prognosis in these patients.

Our study showed an increase in blood cytokine consistent with previous reports [5]. These cytokines might be involved with lung injury. We showed that both lung infiltration and an apoptosis marker increased after renal ischemia reperfusion. Our study confirmed that renal ischemia reperfusion crosstalks with other organs, consistent with previous studies [19–20].

SP-A and SP-D play important roles in innate immunity and in the modulation of inflammation in pulmonary infections [6, 7]. Also serum SP-A and SP-D might be good indicators to reflect the lung injury in many lung diseases [8–10]. But there is no report about the change of serum SP-A and SP-D after AKI. We found that serum SP-A and SP-D increased after renal ischemia reperfusion. According to the result of our study, the expression trend of SP-A and SP-D is different from each other. SP-D responds in latent phase while SP-A has more timely response and increased early after ischemia reperfusion injury. These results are similar to our previously reported study where we showed the changes of SP-A and D after the injection of paraquat [21]. Moreover, SP-A could show an early response to stress like the previous report [22]. In addition, the immunohistochemistry results showed that the expression of these proteins did not change in the lungs. Also, there was no change of these protein levels in the lungs. Although SP-D in BAL is increased at 24 hours, SP-A and RAGE, which are pneumocyte I markers, were not changed after renal ischemia reperfusion. Although SP-A and SP-D are predominantly expressed in the lungs, they are also expressed in other tissues/organs including the kidneys. A recent study showed that lipopolysaccharide can increase SP-A synthesis in human renal epithelial cells through sequentially activating the TLA-4-related MERK1-ERK1/1-NF-κB-dependent pathway [11]. Also, other reports showed that SP-A and SP-D attenuated kidney injury by modulating inflammation and apoptosis [12–14], but they did not investigate the blood level of SP-A and SP-D or lung injury. In our study, changes of SP-A and SP-D levels in the kidneys were similar to the serum changes of these proteins. We think that the serum SP-A level could reflect kidney injury, but SP-D could originate with both the kidneys and the lungs, because the increase of SP-D in BAL resulted from loss of the tight-junction protein.

Also, our results suggested that AKI may increase serum SP-A and SP-D levels; hence AKI should be considered when interpreting the results of SP-A and SP-D in pulmonary disease.

We investigated the lung tight-junction protein because the protein in BAL increased, although there was no damage of pneumocyte I and II. We think these could reflect leakage between the lungs and blood in alveolar space. Also, increased infiltration and apoptosis could be involved with changes of lung architecture. We showed that the tight-junction protein decreased. The apoptosis causes the degradation of the alveolar barrier by down-regulating the expression of tight-junction proteins, especially claudin-4, that inhibit alveolar fluid clearance, In our study we also found the reduced expression of claudin-4, claudin-18, and JAMA-1, which could explain the high incidence of bacteremia incidence in patients with AKI and pneumonia.

In summary, we concluded that serum SP-A and SP-D increased after renal ischemia reperfusion, but these could have originated in the kidneys; so serum SP-A and D could not reflect lung injury in AKI. After renal ischemia reperfusion, lung inflammation and apoptosis increased, which influenced the lung tight-junction protein. Our schematic overview is shown in Fig 7. Further study needs to reveal how tight-junction protein change could influence the prognosis in patients with AKI and pneumonia.

## Supporting information

**S1 Appendix. Supplementary materials and methods.**
(DOCX)

**S1 Fig. Neutrophil infiltration in the lungs after ischemic kidney injury.**
(TIF)

**S1 Raw Images. Original images for western blots.**
(PDF)

**S1 File. ARRIVE Guidelines checklist.**
(PDF)

## Author Contributions

**Conceptualization:** Islam Md Imtiazul, Ho-Yeon Song, Hyo-Wook Gil.

**Formal analysis:** Islam Md Imtiazul, Redwan Asma, Nam-Jun Cho.

**Funding acquisition:** Hyo-Wook Gil.

**Investigation:** Islam Md Imtiazul, Redwan Asma, Ji-Hye Lee, Nam-Jun Cho, Samel Park.

**Methodology:** Islam Md Imtiazul, Samel Park, Ho-Yeon Song, Hyo-Wook Gil.

**Resources:** Ji-Hye Lee, Ho-Yeon Song, Hyo-Wook Gil.

**Supervision:** Ho-Yeon Song, Hyo-Wook Gil.

**Visualization:** Ji-Hye Lee.

**Writing – original draft:** Islam Md Imtiazul.

**Writing – review & editing:** Hyo-Wook Gil.

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
