## [Decision Letter · Decision Letter 0]

31 Oct 2019

PONE-D-19-26349

Change of surfactant protein D and A after renal ischemia reperfusion injury

PLOS ONE

Dear Dr Gil,

Thank you for submitting your manuscript to PLOS ONE. After careful consideration, we feel that it has merit but does not fully meet PLOS ONE’s publication criteria as it currently stands. Therefore, we invite you to submit a revised version of the manuscript that addresses the points raised during the review process.

Your manuscript was reviewed by two experts and  they send mixed responses. All of their comments must be addressed during revision and those are provided below.

We would appreciate receiving your revised manuscript by Dec 15 2019 11:59PM. To enhance the reproducibility of your results, we recommend that if applicable you deposit your laboratory protocols in protocols.io, where a protocol can be assigned its own identifier (DOI) such that it can be cited independently in the future. For instructions see: http://journals.plos.org/plosone/s/submission-guidelines#loc-laboratory-protocols

We look forward to receiving your revised manuscript.

Kind regards,

Partha Mukhopadhyay, Ph.D.

Academic Editor

PLOS ONE

Journal Requirements:

2.  To comply with PLOS ONE submissions requirements, please provide method(s) of sacrifice in the Methods section of your manuscript.'

3.  As part of your revision, please complete and submit a copy of the ARRIVE Guidelines checklist, a document that  aims to improve experimental reporting and reproducibility of animal studies for purposes of post-publication data analysis and reproducibility: https://www.nc3rs.org.uk/arrive-guidelines. Please include your completed checklist as a Supporting Information file. Note that if your paper is accepted for publication, this checklist will be published as part of your article. Please also include the approval number given to your study by your ethics committee.

4. We noticed you have some minor occurrence(s) of overlapping text with the following previous publication(s), which needs to be addressed:

https://doi.org/10.3892/or.2018.6381

https://doi.org/10.1371/journal.pone.0178665

In your revision ensure you cite all your sources (including your own works), and quote or rephrase any duplicated text outside the Methods section. Further consideration is dependent on these concerns being addressed.

Reviewers' comments:

Reviewer's Responses to Questions

**Comments to the Author**

1. Is the manuscript technically sound, and do the data support the conclusions?

Reviewer #1: Partly

Reviewer #2: Partly

2. Has the statistical analysis been performed appropriately and rigorously? 

Reviewer #1: Yes

Reviewer #2: Yes

3. Have the authors made all data underlying the findings in their manuscript fully available?

Reviewer #1: Yes

Reviewer #2: No

4. Is the manuscript presented in an intelligible fashion and written in standard English?

Reviewer #1: Yes

Reviewer #2: Yes

5. Review Comments to the Author

Reviewer #1: The manuscript entitled “Change of surfactant protein D and A after renal ischemia reperfusion injury” suggested a link between serum surfactant protein and renal ischemia reperfusion models. Furthermore, increased lung inflammation and apoptosis were observed due to change of lung tight-junction protein. The authors demonstrated the cross-talk between organs (kidney and lung), however, several concerns must be addressed.

<major concerns="">

- The information in the manuscript does not provide a very well-explainable data between AKI and surfactant protein. Albeit further study will be required in the future, certain conclusion must be given in the manuscript. Therefore, it is pretty tricky to consider the overall ideas the authors provided. The authors need to provide schematic overview of these results for the readers to understand easily.

- There are too much information on materials and methods. The authors need to make it simplify.

- General figure image quality is too poor. It is very difficult to recognize clearly (histology, graph and western blot). It must be addressed to increase image quality (more than 300DPI).

- In results part, the “sub-title” must summarize the results obtained. There is no information of the sub-title the authors provided in the manuscript. It will be re-organized.

- The methods used are described in FIGURE LEGEND. Do not state the results in the FIGURE LEGEND.

For example,

(FIG 1A) Renal function in C57BL/6 mice renal ischemia reperfusion blood urea nitrogen (BUN) and creatinine “increased”  “evaluated”;

(FIG 1B) Tubular injury necrosis scores for PAS-stained kidney sections showed increased tubular injury in ischemic AKI  kidney sections were subjected to PAS-staining and histological changes were scored.

- Fig3B and Fig4 Western blot images, the bands were cut group by group. The authors must provide the entire row immune-blotting bands. Do not cut (or split) the Western bands.

<minor concerns="">

- Figure1A, the order of graphs (BUN and creatine) should be changed. The manuscript stated serum creatinine and BUN (in the results). Try to make the consistency.

- Figure1B, please indicate the areas of tubular damage (i.e. using arrow head)

- Line 182, “protein in BAL increased in the AKI”. Which protein? How to measure? Or any protein changes between shame and AKI models?

- There is no scale bars on Figures1,3 and5 (microscopic images)

- In figure2, put Blood serum on top of the graphs to recognize them easily.

- In figure3A, how to measure neutrophil infiltration? By FACS or by count? If it was evaluated by count, only 1-2 neutrophils in the field? Make them more clarify.

-In figure4 Western blotting, there is no information about the bands. Once again, do not split the western blot images.

Reviewer #2: The authors reported changes of SP-A and SP-D in an AKI mice model at different time points and compared serum and kidney SP-A & SP-D changes to other organ specific markers. They concluded that kidney-originated SP-A/D changes could not be ruled out and thus the two markers are unsuitable for determining AKI-related lung injury. The overall experimental design is adequate, however more details in animal experiments are welcomed. I have a few comments/questions:

In Materials and methods, please indicate sex of animals used.

How was animal sacrifice performed? Spine dislocation or anesthetic overdose? Please specify.

Please define ‘80% return’ in BAL collection paragraph and specify pH or product number of PBS used.

Did you conduct PBS flush of lung vasculature before extracting BAL? why or why not?

Which parts of centrifugated serum or BAL samples were kept? Please specify.

In results:

In SP-A and SP-D assays, error bars are relatively outstanding. Specifically, Lung SP-A protein assay displayed inconsistency in 24h sham group, can the author address the cause of this anomaly? 4hr sham groups and 24h sham group underwent similar surgical procedures, and yet SP-A protein in kidney shows reduction in 24hr sham vs. 4hr sham. Which leads to suspicion that the ‘sham’ surgery is too invasive and disruptive to kidney function, Can the authors please address this phenomenon?

In some data presentation, one group presents much more substantial error bar than other groups. In common knowledge, age and body weight controlled mice are expected to have very consistent biological and biochemical profile under consistent modeling technique. I have to say that the data quality is questionable.

In multiple graphs, SP-A and SP-D shows different trends, it would appear to me that SP-D responds in latent phase while SP-A has more timely response and increased very soon after ischemia reperfusion injury occurs. Why we should look at both targets at the same time, and how do we interpret the difference in their responses? I hope the authors can give a bit discussion in this aspect.

The topic brought up by the authors warrants attention, however the results are of less that satisfying quality. It would be great if well controlled experiments are performed and consistent data is presented, then the same conclusions will be much more convincing.

 </minor></major>

6. PLOS authors have the option to publish the peer review history of their article (what does this mean?). If published, this will include your full peer review and any attached files.

Reviewer #1: No

Reviewer #2: No

---

## [Author Response · Author response to Decision Letter 0]

24 Nov 2019

Dear Partha Mukhopadhyay, Ph.D.

Thank you so much for your comments. I am glad to see your comments, indicating that I have a chance to revise my manuscript. Following the reviewer’s comments, I have amended my manuscript as written below. In text, the changes and corrections have been highlighted by yellow color and green color changes mean deleted sentences. I hope you would review it again and I hope for the honor of accepting my scientific writing in your highly authorized journal.

Reviewer 1 comments

Reviewer #1: The manuscript entitled “Change of surfactant protein D and A after renal ischemia reperfusion injury” suggested a link between serum surfactant protein and renal ischemia reperfusion models. Furthermore, increased lung inflammation and apoptosis were observed due to change of lung tight-junction protein. The authors demonstrated the crosstalk between organs (kidney and lung), however, several concerns must be addressed.

Comment 1. The information in the manuscript does not provide a very well-explainable data between AKI and surfactant protein. Albeit further study will be required in the future, certain conclusion must be given in the manuscript. Therefore, it is pretty tricky to consider the overall ideas the authors provided. The authors need to provide schematic overview of these results for the readers to understand easily.

Answer: Thank you for your kind suggestion. A schematic overview has been added in the revised manuscript. Kindly, check the figure 7 in the revised manuscript.

Comment 2. There is too much information on materials and methods. The authors need to make it simplify.

Answer: We have simplified the materials and methods according to the suggestion in the revised manuscript. Some details of materials and methods have been transferred in supplementary document. 

Comment 3. General figure image quality is too poor. It is very difficult to recognize clearly (histology, graph and western blot). It must be addressed to increase image quality (more than 300DPI).

Answer: We have improved the resolution of all figures. The image quality has been increased according to the suggestion in the revised manuscript.

Comment 4. In results part, the “sub-title” must summarize the results obtained. There is no information of the sub-title the authors provided in the manuscript. It will be re-organized.

Answer: The subtitles of the result part have been reorganized and the summarized result has been added to the result subtitles of the revised manuscript. 

Comment 5. The methods used are described in FIGURE LEGEND. Do not state the results in the FIGURE LEGEND.

For example,

(FIG 1A) Renal function in C57BL/6 mice renal ischemia reperfusion blood urea nitrogen (BUN) and creatinine “increased”  “evaluated”;

(FIG 1B) Tubular injury necrosis scores for PAS-stained kidney sections showed increased tubular injury in ischemic AKI  kidney sections were subjected to PAS-staining and histological changes were scored.

Answer: The figure legends have been changed according to your comment. 

Comment 6. Fig 3B and Fig4 Western blot images, the bands were cut group by group. The authors must provide the entire row immune-blotting bands. Do not cut (or split) the Western bands.

Answer: Thank you for the suggestion and the entire row immune-blotting bands has been provided according to the suggestion in the revised manuscript. When we did western blot, the band order is 24hr sham, 24hr AKI, 4hr Sham, and 4hr AKI. I am afraid that these band order make readers confused. 

Comment 7. Figure1A, the order of graphs (BUN and creatine) should be changed. The manuscript stated serum creatinine and BUN (in the results). Try to make the consistency.

Answer: Thank you for the observation. The correction has been done in the revised manuscript.

Comment 8. Figure1B, please indicate the areas of tubular damage (i.e. using arrowhead)

Answer: Thank you for the suggestion. The arrow has been provided to indicate the areas of tubular damage in the figure 1B in the revised manuscript.

Comment 9. Line 182, “protein in BAL increased in the AKI”. Which protein? How to measure? Or any protein changes between shame and AKI models?

Answer: In previous manuscript “protein in BAL increased in the AKI” meant the “total protein” in BAL increased in AKI compared to sham. The total protein in the BAL of all sham and AKI models was measured by using Pierce BCA Protein Assay Kit (Thermo Fisher Scientific). The result showed that the amount of total protein in BAL was higher in AKI than sham.

Comment 10. There is no scale bars on Figures1, 3 and 5 (microscopic images)

Answer: The scale bars have been inserted. 

Comment 11. In figure 2, put Blood serum on top of the graphs to recognize them easily.

Answer: The modification has been done in the revised manuscript according to the suggestion.

Comment 12. In figure 3A, how to measure neutrophil infiltration? By FACS or by count? If it was evaluated by count, only 1-2 neutrophils in the field? Make them more clarify.

Answer: Yes, we measured the neutrophil infiltration by counting and the count is correct. The pathologist scored the neutrophil count on a 400-fold slide as follows.

0 (rare neutrophils in capillary lumen), 1 (frequent neutrophils in capillary lumen), 2 (extravasation of neutrophils), 3 (aggregation of neutrophils in alveolar wall).

Comment 13. In figure 4 Western blotting, there is no information about the bands. Once again, do not split the western blot images.

Answer: Thank you for showing us the mistake and we believe it will help us to improve the quality of our article. The information and whole western blot picture have been added to the revised manuscript.

Reviewer 2 comments

Reviewer #2: The authors reported changes of SP-A and SP-D in an AKI mice model at different time points and compared serum and kidney SP-A & SP-D changes to other organ specific markers. They concluded that kidney-originated SP-A/D changes could not be ruled out and thus the two markers are unsuitable for determining AKI-related lung injury. The overall experimental design is adequate, however more details in animal experiments are welcomed. I have a few comments/questions:

Comment 1. In Materials and methods, please indicate sex of animals used.

Answer: In the revised manuscript we have added the sex (male) of the animals.

Comment 2. How was animal sacrifice performed? Spine dislocation or anesthetic overdose? Please specify.

Answer: We sacrificed our animals by spine dislocation method. In the revised manuscript the information is specified, kindly, find the added information in the revised manuscript.

Comment 3. Please define ‘80% return’ in BAL collection paragraph and specify pH or product number of PBS used.

Answer: Thank you for the question and it had helped us to solve the typing mistake. Previously we mistakenly wrote samples with 80% return were discarded instead of samples with �80% return was discarded. Normally, BAL contains different biochemical and cytological indicators that represent the status of whole lung but the BAL samples that are obtained from �80% return of lavage cannot exactly represent the whole lung status, so we discarded all the samples with �80% return of lavage. 

In the revised manuscript we have corrected the information from 80% to �80% as well as specified the pH (7.4) and the product information (Gibco, USA) for the PBS. 

Comment 4. Did you conduct PBS flush of lung vasculature before extracting BAL? why or why not?

Answer: We did not conduct PBS flush. We think that PBS flush could influence the tight junction barrier because of flush pressure. Also, we did follow previous reports (Kidney Int. 2017 Aug;92(2):365-376, Kidney Int. 2017 May;91(5):1057-1069.)

Comment 5. Which parts of centrifugated serum or BAL samples were kept? Please specify.

Answer: We stored the supernatant of the centrifuged serum and BAL. This information is specified in the revised manuscript. 

Comment 6. In results:

In SP-A and SP-D assays, error bars are relatively outstanding. Specifically, Lung SP-A protein assay displayed inconsistency in 24h sham group, can the author address the cause of this anomaly? 4hr sham groups and 24h sham group underwent similar surgical procedures, and yet SP-A protein in kidney shows reduction in 24hr sham vs. 4hr sham. Which leads to suspicion that the ‘sham’ surgery is too invasive and disruptive to kidney function, Can the authors please address this phenomenon?

Answer: When we made the sham group, we tried to follow the same procedure as AKI model except for ischemic reperfusion. I have already published that Sham itself affect heart metabolites (Kidney Int. 2019 Mar;95(3):590-610). Although we had made effort to be less invasive when making sham, but sham operation itself could be engaged with injury. BUN, Creatinine and kidney histology are stable in both two shams. 

Comment 7. In results:

In some data presentation, one group presents much more substantial error bar than other groups. In common knowledge, age and body weight-controlled mice are expected to have very consistent biological and biochemical profile under consistent modeling technique. I have to say that the data quality is questionable.

Answer: BUN/Cr showed similar values in same groups. But some surfactant protein values have wide range. This might be an individual difference in response to IR, but it is thought that this may be a wide range because the amount (pg/ml) of surfactant protein is very small compared to bun/cr (mg/dl). 

Comment 8. In results:

In multiple graphs, SP-A and SP-D shows different trends, it would appear to me that SP-D responds in latent phase while SP-A has more timely response and increased very soon after ischemia reperfusion injury occurs. Why we should look at both targets at the same time, and how do we interpret the difference in their responses? I hope the authors can give a bit discussion in this aspect.

Answer: Thank you for your excellent comment. We checked surfactant protein A and D change after paraquat injection. (Korean J Intern Med. 2007 Jun;22(2):67-72.) The current pattern is similar with this study. I agree that SP-A has early response to stress (Lab Invest. 1996 Jan;74(1):209-20.). We have incorporated these in discussion section. 

Comment 9. The topic brought up by the authors warrants attention, however the results are of less that satisfying quality. It would be great if well controlled experiments are performed and consistent data is presented, then the same conclusions will be much more convincing.

Answer: Thank you for the valuable comment. I think that further study needs more numbers of animals in each group and should uncover the role of surfactant protein in each specific organ. For the future direction, we will design experimental study to prove the crosstalk of surfactant proteins between kidney and lung in mice and validate this employing human data.

---

## [Decision Letter · Decision Letter 1]

6 Dec 2019

PONE-D-19-26349R1

Change of surfactant protein D and A after renal ischemia reperfusion injury

PLOS ONE

Dear Dr Gil,

Thank you for submitting your manuscript to PLOS ONE. After careful consideration, we feel that it has merit but does not fully meet PLOS ONE’s publication criteria as it currently stands. Therefore, we invite you to submit a revised version of the manuscript that addresses the points raised during the review process.

Your manuscript was reviewed by same reviewers and we received positive feedback from them. However, numerous errors and minor technical questions raised by one of the reviewers.

We would appreciate receiving your revised manuscript by Jan 20 2020 11:59PM. To enhance the reproducibility of your results, we recommend that if applicable you deposit your laboratory protocols in protocols.io, where a protocol can be assigned its own identifier (DOI) such that it can be cited independently in the future. For instructions see: http://journals.plos.org/plosone/s/submission-guidelines#loc-laboratory-protocols

We look forward to receiving your revised manuscript.

Kind regards,

Partha Mukhopadhyay, Ph.D.

Academic Editor

PLOS ONE

Reviewers' comments:

Reviewer's Responses to Questions

**Comments to the Author**

1. If the authors have adequately addressed your comments raised in a previous round of review and you feel that this manuscript is now acceptable for publication, you may indicate that here to bypass the “Comments to the Author” section, enter your conflict of interest statement in the “Confidential to Editor” section, and submit your "Accept" recommendation.

Reviewer #1: (No Response)

Reviewer #2: All comments have been addressed

2. Is the manuscript technically sound, and do the data support the conclusions?

Reviewer #1: Yes

Reviewer #2: Yes

3. Has the statistical analysis been performed appropriately and rigorously? 

Reviewer #1: Yes

Reviewer #2: Yes

4. Have the authors made all data underlying the findings in their manuscript fully available?

Reviewer #1: Yes

Reviewer #2: Yes

5. Is the manuscript presented in an intelligible fashion and written in standard English?

Reviewer #1: Yes

Reviewer #2: Yes

6. Review Comments to the Author

Reviewer #1: 1. In Fig1A, Y axis of the graphs, mg/dL and mg/dl are mixed. Make them consistent.

2. In the manuscript, put the full name of ATN (might be “Acute tubular necrosis”), related with Fig1B

3. In Fig3A, H&E staining for neutrophil is not clear to determine NP infiltration. The evaluation of NP infiltration was done by pathologist according to the authors explanation and it was fully understandable. However, please perform MPO staining to confirm NP infiltration on those sites and put that data on Supporting figure?

4. In Fig4 (western blot), JAMA-1 or JAM-A? clarify

5. In Fig4, Claudin 3 , Claudin 4, Claudin 18 and Claudin-3, Claudin-4, Claudin-18 are mixed. Make them consistent.

6. In Fig6, the staining SP-A and SP-D was not still clear. Please count SP-A and SP-D positive cells and add one other graph how many positive cells were counted in each group.

Reviewer #2: The authors has included additional materials supplementing the findings and completing the data presentations. previous concerns are addressed by additional descriptions or discussion.

7. PLOS authors have the option to publish the peer review history of their article (what does this mean?). If published, this will include your full peer review and any attached files.

Reviewer #1: No

Reviewer #2: No

---

## [Author Response · Author response to Decision Letter 1]

11 Dec 2019

Dear Partha Mukhopadhyay, Ph.D.

Thank you so much for your comments. I am glad to see your comments, indicating that I have a chance to revise my manuscript. Following the reviewer’s comments, I have amended my manuscript as written below. In text, the changes and corrections have been highlighted by yellow color and green color changes mean deleted sentences. I hope you would review it again and I hope for the honor of accepting my scientific writing in your highly authorized journal.

Reviewer 1 comments

Comment 1. In Fig 1A, Y axis of the graphs, mg/dL and mg/dl are mixed. Make them consistent.

Answer: Thank you for showing us the mistake and we hope it will help us to upgrade the quality of our manuscript. The correction has been completed and we have made mg/dl consistent in Fig 1A in the revised manuscript.

Comment 2. In the manuscript, put the full name of ATN (might be “Acute tubular necrosis”), related with Fig 1B 

Answer: Thank you for the suggestion. We have added the full name of Acute tubular necrosis (ATN) in the revised manuscript. Kindly check the revised manuscript, page 6, line 132 and 138.

Comment 3. In Fig 3A, H&E staining for neutrophil is not clear to determine NP infiltration. The evaluation of NP infiltration was done by pathologist according to the authors explanation and it was fully understandable. However, please perform MPO staining to confirm NP infiltration on those sites and put that data on Supporting figure? 

Answer: Thank you for the good comment. I agree with your point. Unfortunately, there is a problem with the block, which is not suitable for MPO staining. Considering what we can technically do, we took the picture by x 1000 times and added as a supplementary figure in the revised manuscript. Kindly, check the supplementary Fig S1 in the supplementary document and additionally we have provided the picture below: 

Comment 4. In Fig 4 (western blot), JAMA-1 or JAM-A? clarify

Answer: Thank you for showing us the mistake. In the Fig 4 it will be JAMA-1 (anti-junctional adhesion molecule 1) and we have corrected the Fig 4 in the revised manuscript. 

Comment 5. In Fig 4, Claudin 3, Claudin 4, Claudin 18 and Claudin-3, Claudin-4, Claudin-18 are mixed. Make them consistent.

Answer: Thank you for the suggestion. In the Fig 4, the correction has been completed according to the reviewer’s comment. 

Comment 6. In Fig 6, the staining SP-A and SP-D was not still clear. Please count SP-A and SP-D positive cells and add one other graph how many positive cells were counted in each group.

Answer: Thank you for the suggestion. We have counted the SP-A and SP-D positive cells per HPF. We also insert the additional graph in Fig 6 in the revised manuscript.

---

## [Editor Report · Decision Letter 2]

13 Dec 2019

Change of surfactant protein D and A after renal ischemia reperfusion injury

PONE-D-19-26349R2

Dear Dr. Gil,

We are pleased to inform you that your manuscript has been judged scientifically suitable for publication and will be formally accepted for publication once it complies with all outstanding technical requirements.

With kind regards,

Partha Mukhopadhyay, Ph.D.

Section Editor

PLOS ONE
---

## [Editor Report · Acceptance letter]

17 Dec 2019

PONE-D-19-26349R2 

Change of surfactant protein D and A after renal ischemia reperfusion injury 

Dear Dr. Gil:

I am pleased to inform you that your manuscript has been deemed suitable for publication in PLOS ONE. Congratulations! Your manuscript is now with our production department. 

With kind regards,

on behalf of

Dr. Partha Mukhopadhyay 

Section Editor

PLOS ONE